# Evaluation of Logistics-Industry Efficiency and Enhancement Path in China's Yellow River Basin under Dual Carbon Targets

Changrong Dong [1,*], Gongmin Zhao [1], Yuanhao Wang [2], Yongjie Wu [1] and Haimeng Liang [1]

1   School of Economics and Management, North University of China, Taiyuan 030051, China
2   Twenty-Fifth Institute of the Second Academy, China Aerospace Science & Industry Corporation, Beijing 100854, China
*   Correspondence: d1378021048@163.com; Tel.: +86-158-3415-1458

**Abstract:** Enhancing circulation efficiency and helping green development have become essential requirements for ecological protection and high-quality development in the Yellow River Basin. Based on the panel data of China's Yellow River Basin from 2011 to 2020, the static and dynamic efficiencies of the logistics industry in the nine provinces of the Yellow River Basin were measured by using the super-efficiency SBM model and the Malmquist index model. Then, government support, economic level, industrial agglomeration, technological innovation, openness, and environmental regulation were selected as antecedent variables; the efficiency value of the logistics industry in the Yellow River Basin was selected as the outcome variable; and combined with the fsQCA method, analyzed from a group perspective, four group paths were obtained: economic openness-type path, technology industry-type path, government intervention-type path, and non-environmental-regulation path. The conclusion can deepen the systematic understanding of the group path of the logistics industry and can provide theoretical guidance for regions to improve the efficiency of the logistics industry and promote green development.

**Keywords:** efficiency of the logistics industry; super-efficient SBM; Malmquist model; dual carbon perspective; fsQCA

## 1. Introduction

On 16 October 2022, the report of the 20th Congress of the Communist Party of China (a meeting to study and decide the most important issues of the Communist Party of China) clearly pointed out that it was necessary to accelerate the construction of a new development pattern and focus on promoting high-quality development. The construction of an efficient and smooth circulation system, reducing logistics costs, and improving the efficiency of the logistics industry are of great significance to promoting high-quality economic development. For a long time, the logistics industry, as a basic and strategic industry in China's national economic development, has had obvious characteristics of crude growth, with high input, high consumption, and low output [1]. As a major carbon source, the energy consumption of China's logistics industry has now exceeded 20% of the total energy consumption of the whole society, and the $CO_2$ emissions of the logistics industry reached 790 million tons in 2020, accounting for 8% of the proportion of the whole society and 70% of the proportion of the service industry, which is one of the fastest growing industries in China in terms of carbon emissions. The low efficiency of the logistics industry has become a major challenge to the implementation of the "double carbon goal" and high-quality development. Thus, it is clear that for the protection of economic growth without slippage, the scientific and effective reduction of the carbon-emission growth rate of the logistics industry, and constantly improving the efficiency of the logistics industry, the sustainable development of China's future economic and social development is of

great significance. This is especially true as China moves from its previous pursuit of economic efficiency using the rough development model to the pursuit of a stable quantity of quality development.

The adoption of the Outline of the Plan for Ecological Protection and High-Quality Development of the Yellow River Basin and the Law of the People's Republic of China on the Protection of the Yellow River in 2021 has brought the high-quality development of the Yellow River Basin to a new level. Under the dual requirements of a "double carbon" target and high-quality development, the development pattern of the new era has put forward higher requirements for the development of all industries and regions in China, and the logistics industry has become an important pillar industry under the requirements of high-quality development because of its basic and strategic characteristics. The Yellow River Basin, as an important area for economic development in the north of China and the focus of ecological protection and high-quality development in the current and future periods, also bears the heavy responsibility of achieving the second-century goal of China. Therefore, exploring the path to improve the efficiency of the logistics industry in the Yellow River Basin while considering the reduction of carbon emissions has become a key issue to promote high-quality economic development and green sustainability.

## 2. Literature Review

The numerous studies on the efficiency of the logistics industry in today's academic world evidence that this is one of the hot topics that scholars continue to focus on. In terms of efficiency evaluation methods in the logistics industry, the most applied method is data envelopment analysis (DEA). In terms of research models, it can be divided into three main cases; the first one is a single traditional DEA model. Foreign scholar Schinnar [2] first applied DEA to logistics-efficiency evaluation in 1980, and domestic scholars Jun Zhicai et al. [3] first used DEA to analyze the level of logistics development at that time (in 1994). After that, more and more scholars used the DEA model to analyze and measure the efficiency of the logistics industry. Liu Cong and Li Zhenzhen [4] used the DEA-BCC model to measure the efficiency of the low-carbon logistics industry based on the data of three provinces and one city in the Yangtze River Delta from 2008 to 2019 and found that the efficiency of low-carbon logistics in Yangtze River Delta was generally good and could improve the level of regional economic development. The second type of DEA model is a phased study, whose core objective is to ensure the accuracy of the model measurement. In order to solve the "black box" problem in logistics-efficiency measurement, Zhang Hao and You Jianxin [5] constructed a two-stage logistics-efficiency assessment model based on the DEA model, taking the two stages of investment transformation and social services in the production and operation of the logistics industry as the dividing line. They found that the reasons for the inefficiency of the logistics industry in most provinces are inefficient transformation and inefficient social services. Gong Ruifeng et al. [6] excluded the interference of random factors through a three-stage DEA model and proposed that the key to the improvement of regional logistics efficiency lies in the improvement of scale efficiency. The third is a DEA model used in combination with other research methods, whose main advantage is to make up for the shortcomings of a single model. Xue Gong and Linbo Jing [7] combined the DEA model with the Malmquist index model to measure the static efficiency and dynamic efficiency of China's logistics industry, respectively, and analyzed the dynamic evolution process of China's logistics-industry-efficiency development in terms of comprehensive efficiency and decomposition. However, Rita et al. [8] combined the DEA model with AHP to quantify the efficiency of the logistics industry in 29 European countries more systematically and comprehensively, which made the model more reasonable and convincing.

In terms of the scope of research on the evaluation of logistics-industry efficiency, scholars have mostly analyzed the evaluation of logistics-industry efficiency at the national, regional, provincial, enterprise, or port levels. Liu Huajun et al. [9] analyzed the spatial and temporal pattern of China's logistics-industry efficiency at the national level with sample

data from 30 provinces in China and found that the current efficiency level of China's logistics industry is generally low; there are significant differences between regions; and the efficiency level in the eastern region is much higher than that in other regions. Zheng Jin'e et al. [10], on the other hand, studied the variability of the comprehensive efficiency and decomposition terms of the logistics industry in the Yangtze River Economic Belt at the regional level. In addition, other scholars have evaluated and studied the efficiency of the logistics industry in the Beijing–Tianjin–Hebei region [11], the western region [12], the Changzhutan region [13], and the Guangdong–Hong Kong–Macao Greater Bay Area [14] in China. Based on the provincial level, Zhang Yongsheng [15] evaluated the efficiency of the logistics industry in Guangxi Province and analyzed its influencing factors. At the enterprise or port level, Chu Yanchang et al. [16] studied the logistics efficiency and influencing factors of China's listed companies and found that the overall logistics efficiency of China's listed companies was on the rise and we should pay attention to the introduction of talents and technological innovation. Moreover, Jing Wang and Jia Zhou [17] analyzed the logistics efficiency of 11 ports in the Guangdong–Hong Kong–Macao Greater Bay Area and concluded that the overall logistics efficiency was not high, and there were issues such as inefficient management and homogenization, as well as other problems.

In terms of the factors influencing the efficiency of the logistics industry, relevant studies by domestic and foreign scholars have gradually matured, mainly involving various factors such as the level of economic development, fixed investment amount, information technology level, industrial structure, personnel quality, innovation capability, and environmental governance [6–13]. Foreign scholars' studies on the factors influencing the efficiency of the logistics industry are more placed at the micro level, focusing on how enterprises can improve logistics efficiency in the operation process. Klumpp et al. [18] used the Network DEA method to explore the impact of information technology disruptions on logistics efficiency in the retail industry, emphasizing the improvement of the sustainability and resilience of retail logistics. Viet et al. [19] used the consistent fuzzy-preference relationship approach to analyze the influencing factors of Vietnam's logistics system and made recommendations in terms of investment sequencing and adaptation regulations. Domestic scholars are divided in their understanding of the factors influencing the efficiency of the logistics industry. Yu Lijing and Chen Zhongquan [20] found that government support and scientific and technological progress have a favorable impact on logistics efficiency, while the intensity of environmental regulation, the level of economic development, and openness to the outside world fail to have a positive impact. Liu Chengliang and Guan Mingming [21] found that economic development, market environment, industrial agglomeration, informatization level, and government regulation had a significant positive impact on logistics efficiency, while energy intensity had a negative impact, and the degree of opening up to the outside world and environmental regulation had an insignificant effect on the improvement of logistics efficiency. Zhang Yunfeng and Wang Yu [22] in the study of logistics-industry efficiency factors, found that low carbon constraints and government support have a certain inhibitory effect on the efficiency of the logistics industry, and the level of economic development and the industrial restructuring of the logistics-industry efficiency has a promotional effect. The empirical results of the above literature indicate that the impact of the same factors is different and shows an inconsistent direction, which is, to some extent, closely related to the selection of data, variable selection, research methodology, and so on. The causes or conditions of social phenomena are often interdependent rather than independent, and the independent variables are often multicollinear because of their interrelationships, which means that the unique effects of individual variables may be masked by related variables. Therefore, it is more appropriate to take a holistic and combinatorial approach to explain the occurrence of social phenomena; especially for the multiple concurrency of antecedents, it is suitable to explore the corresponding enhancement path from the group perspective [23].

In summary, the current academic community has, relatively, thoroughly researched the efficiency of the logistics industry in terms of the methodology and scope of the

influencing factors, but there is less analysis of $CO_2$ emissions as non-desired outputs in conjunction with the "double carbon" target. Moreover, the measurement of $CO_2$ emissions also mostly uses raw coal, and there is less use of oil products, natural gas, and other fuels. The use of fuels such as oil and natural gas is less often taken into account. At the same time, most studies on the evaluation of regional logistics-industry efficiency focus on economically developed regions, such as Beijing, Tianjin and Hebei, Guangdong, Hong Kong and Macao, and the Yangtze River Economic Belt, while the Yellow River Basin is not prominent enough in the national economy. However, it is an important part of China's sustainable development and dual carbon goals. In addition, the existing literature, when analyzing the factors influencing the efficiency of the logistics industry, is usually limited to the binary relationship between the independent and dependent variables, while in actual operation, the efficiency level of the logistics industry is often influenced by several factors together, and the different focus or combination of influencing factors will have different strengths on the efficiency value. The study of the strength of a factor alone is not enough to systematically summarize a suitable improvement path. Therefore, from the perspective of "double carbon", this paper uses labor, capital, and energy consumption as input indicators; the comprehensive transportation volume and added value of logistics industry as desired output indicators; and carbon dioxide emissions as non-desired output indicators and applies the super-efficiency SBM model and Malmquist index model to measure the efficiency values of logistics industry in the Yellow River Basin from 2011 to 2020. The efficiency values of the logistics industry in nine provinces from 2011 to 2020 are measured, and the trends of static efficiency and dynamic efficiency changes in different periods and regions are analyzed in detail. After this, the fuzzy-set qualitative comparative analysis (fsQCA) method is selected to study the relationship between the outcome variables and the combination of elements of the condition variables in depth, adopting a holistic perspective and group thinking. On this basis, a path for improving the efficiency of the high-level logistics industry in the Yellow River Basin is proposed. Overall, this paper enriches the research on the efficiency of the logistics industry in the Yellow River Basin, which has certain significance for later scholars, and formulates a development path through a combination of strategies in order to provide a useful reference for the green transformation development of China's logistics industry, the scientific and reasonable planning and layout of governments at all levels, and the early realization of the goal of "double carbon".

## 3. Research Design

### 3.1. Research Methodology

3.1.1. Super-Efficient SBM Model Considering Non-Desired Output

At present, domestic and foreign scholars' research on the efficiency of the logistics industry mainly adopts the data envelopment analysis (DEA) method. In order to overcome the problem that traditional DEA models cannot consider slack variables, some scholars also choose the SBM model proposed by Tone [24] to measure the efficiency of the logistics industry. Although the SBM model simultaneously considers both inputs and outputs, and the two are jointly improved to be effective, circumventing the ineffective solution while also being more in line with the actual situation, the range of measured efficiency values can only be between 0 and 1, and when there are multiple effective units, the SBM model is unable to further evaluate them. In view of this, this paper will use the super-efficient SBM model improved by Tone [25] considering non-expected outputs. In this way, comparisons can also be made between effective units whose original efficiency values are all 1. And carbon dioxide emissions as non-expected outputs are included in the model to measure

the efficiency of the logistics industry in the Yellow River Basin. The improved model is shown in Equation (1).

$$\theta^* = min \frac{1 + \left[\frac{1}{L}\sum_{l=1}^{L}\frac{s_l^-}{x_{l,o}^t}\right]}{1 - \left[\frac{1}{M+N}\left(\sum_{m=1}^{M}\frac{s_m^+}{y_{m,o}^t} + \sum_{n=1}^{N}\frac{s_n^-}{z_{n,o}^t}\right)\right]}$$

$$s.t. \begin{cases} \sum_{t=1}^{T}\sum_{i=1,i\neq o}^{I} a_i^t x_{i,l}^t - s_l^- \leq x_{l,o}^t, l = 1,2,\cdots,L \\ \sum_{t=1}^{T}\sum_{i=1,i\neq o}^{I} a_i^t y_{i,m}^t + s_m^+ \geq y_{m,o}^t, m = 1,2,\cdots,M \\ \sum_{t=1}^{T}\sum_{i=1,i\neq o}^{I} a_i^t z_{i,n}^t - s_n^- \leq z_{n,o}^t, n = 1,2,\cdots,N \\ a_i^t \geq 0, s_l^- \geq 0, s_m^+ \geq 0, s_n^- \geq 0, i = 1,2,\cdots,I \end{cases}$$

(1)

In Equation (1), $\theta^*$ denotes the efficiency value; $L$, $M$, and $N$ denote the number of logistics-industry input indicators, desired output indicators, and non-desired output indicators, respectively; $l = [1, L]$, $m = [1, M]$, and $n = [1, N]$; $s_l^-$, $s_m^+$ and $s_n^-$ denote the slack vectors of logistics-industry input indicators, desired output indicators, and non-desired output indicators, respectively; $x_{i,l}^t$, $y_{i,m}^t$ and $z_{i,n}^t$ denote the input-output values of the decision unit in period t, respectively; $x_{l,o}^t$, $y_{m,o}^t$ and $z_{n,o}^t$ denote the input-output variables of the o decision unit; and $a_i^t$ denote the weights of that decision unit.

### 3.1.2. Global Malmquist Index Model

DEA models have the advantages of no data processing and no division of weights, and they are widely used by scholars because of their convenience and scientific nature. However, DEA models can only measure the efficiency value of a decision unit at a certain time, i.e., static efficiency, and cannot measure the technical changes of the decision unit in different periods and the resulting movement of the production frontier, i.e., dynamic efficiency [26]. Therefore, this paper will adopt the Malmquist–Luenberger index [27,28], which is an effective and widely used global reference, to measure the total factor productivity of the logistics industry in the Yellow River Basin, as well as the decomposition of the technical efficiency index (*EC*) and the technological progress index (*TC*). This will allow us to dynamically understand the trend of the efficiency changes and trace the reasons for the changes in the efficiency, at the same time solving the problem of the lack of feasible solutions. When *ML* > 1, it means that the productivity of the decision-making unit is increasing; when *ML* < 1, it means that the productivity is decreasing; the *EC* index and *TC* index are the same; the *ML* index model is shown in Equation (2).

$$(ML)_t^{t+1} = \left[\frac{D_v^t(x^{t+1},y^{t+1},z^{t+1})}{D_v^t(x^t,y^t,z^t)} \times \frac{D_v^{t+1}(x^{t+1},y^{t+1},z^{t+1})}{D_v^{t+1}(x^t,y^t,z^t)}\right]^{\frac{1}{2}}$$

$$= \frac{D_v^{t+1}(x^{t+1},y^{t+1},z^{t+1})}{D_v^t(x^t,y^t,z^t)} \times \left[\frac{D_v^t(x^{t+1},y^{t+1},z^{t+1})}{D_v^{t+1}(x^{t+1},y^{t+1},z^{t+1})} \times \frac{D_v^t(x^t,y^t,z^t)}{D_v^{t+1}(x^t,y^t,z^t)}\right]^{\frac{1}{2}}$$

(2)

$$h = EC_t^{t+1} \times TC_t^{t+1}$$

In Equation (2), $x^t$, $y^t$ and $x^{t+1}$, $y^{t+1}$ denote the input and output in period $t$ and period $t + 1$, respectively; $z^t$, $z^{t+1}$ denote the non-desired output; $D^t$ and $D^{t+1}$ denote the distance function.

### 3.1.3. Qualitative Comparative Analysis of Fuzzy-Set fsQCA

Fuzzy-set qualitative comparative analysis (fsQCA) is a method for exploring the combinations of antecedent conditions that lead to the production of a particular outcome [29], thereby explaining the complex causal relationships behind the phenomenon. This paper adopts the fsQCA method from a holistic and group thinking perspective, mainly based

on the following considerations: First, it is free from the limitations of the traditional regression analysis-based quantitative research focusing on the binary relationship between "independent variables—dependent variables" and adopts multiple case studies to avoid the drawbacks of single analysis. Additionally, it is suitable for exploring the multi-factor parallel causality and multi-factor synergistic mechanism of logistics-industry efficiency in the Yellow River Basin. Second, the fsQCA method considers that the same result can be generated by different paths, which is more in line with the actual high-quality development of the logistics industry in the Yellow River Basin of China. Third, the fsQCA method combines the advantages of qualitative and quantitative analysis and is mainly used for the analysis and research of small and medium samples, while the data of nine provinces in the Yellow River Basin belong to small sample cases, which fits well with the needs of this study. The fsQCA approach assesses the relationship between the explanatory variable ($X$) and the explanatory variable ($Y$) through consistency and coverage. Consistency refers to the proportion of all cases on the path that reach the target threshold level, indicating the degree of belief in the path, while coverage indicates the number of cases that follow the path as a proportion of the total, reflecting the explanatory power of the model. It is generally accepted that $X$ is necessary for $Y$ when the consistency > 0.9, and the related formula is as follows:

$$Consistency(X \rightarrow 1) = \frac{\sum \min(X(i), Y(i))}{\sum X(i)} \tag{3}$$

$$Coverage(X \rightarrow 1) = \frac{\sum \min(X(i), Y(i))}{\sum Y(i)} \tag{4}$$

*3.2. Selection of Indicator Variables and Data Sources*

3.2.1. Logistics-Industry Efficiency Measurement Indicators

Regarding the selection of efficiency evaluation indicators for the logistics industry, previous scholars [4–7] have mostly selected input indicators from human, material, and financial resources, and output indicators from economic and operational aspects. In this paper, we refer to the existing research [4–11], adhering to the principles of scientific comprehensiveness and accessibility, and based on the perspective of "double carbon", we also include carbon dioxide emissions of non-desired outputs in the evaluation index system to obtain the input and output indicators of logistics-industry efficiency in the Yellow River Basin (as shown in Table 1).

**Table 1.** Yellow River Basin logistics-industry efficiency input-output indicators.

| Indicator Type | | Specific Indicators |
|---|---|---|
| Input Indicators | Capital inputs | Investment in fixed assets in logistics industry (RMB billion) |
| | Labor input | Number of logistics employees (people) |
| | Energy input | Energy consumption (million tons of standard coal) |
| Output Indicators | Desired output | Value added of logistics industry (RMB billion) |
| | | Comprehensive turnover (billion ton kilometers) |
| | Undesired outputs | $CO_2$ emissions (million tons) |

At present, there are no direct statistics of the logistics industry in China, and according to the China Logistics Statistical Yearbook, 85% of the added value of China's logistics industry comes from the transportation, storage, and postal industries, which are the main representatives. Thus, this paper selects the data of the transportation, storage, and postal industries instead of the logistics industry by referring to Liu Huajun et al. [9]. The time period selected in this paper is 2011–2020, and the nine provinces through which the Yellow River flows are identified as the Yellow River Basin in existing studies [30], which is divided into the upper, middle, and lower reaches with reference to general practice. Among them, the upstream includes Qinghai, Sichuan, Gansu, and Ningxia provinces; the midstream

includes Inner Mongolia, Shaanxi, and Shanxi provinces; and the downstream includes Henan and Shandong provinces. All raw data involved in the study were obtained from China Statistical Yearbook, China Logistics Statistical Yearbook, China Energy Statistical Yearbook, etc., and the relevant data were processed as necessary.

When considering the capital input in the logistics industry, the current estimation of capital stock is dominated by the perpetual inventory method [31], so we refer to the practice of Yunning Zhang et al. [32], replacing it with the fixed asset investment in the logistics industry and converting it to the comparable price in 2011. When considering labor input, the number of employees in the logistics industry at the end of the year is used as a proxy. When considering energy inputs in the logistics industry, the existing studies mainly focus on raw coal and gasoline, but in this paper, based on the "double carbon" perspective, we select the top seven energy sources consumed in the logistics industry (see Table 2) and convert them into the form of standard coal according to the conversion factor. When considering the expected output, the GDP deflator is used to deflate the value added of the logistics industry, using 2011 as the base period, to further eliminate price effects. The passenger turnover and freight turnover cannot be added directly, but after conversion according to the conversion ratio in the "Cost Management and Accounting Measures for Transportation Enterprises" (1 person-km = 1 ton-km by rail = 1 ton-km by waterway, 10 person-km = 1 ton-km by road), the integrated turnover is obtained, which integrates the transportation volume and distance while eliminating the price limitation. This provides a good evaluation of the output level of logistics operations. When considering the non-desired output, the literature has mostly adopted the method proposed in the 2006 IPCC Guidelines for National Greenhouse Gas Inventories to measure the carbon emissions of each province [9], and the integrated carbon dioxide emissions of the logistics industry are calculated as in Equation (5). The conversion factors of the seven main energy sources consumed are shown in Table 2.

$$CO_2 = \sum_{i=1}^{n} E_i \times CF_i \times CC_i \times COF_i \times (44 \div 12) \tag{5}$$

**Table 2.** The seven main energy-conversion standards for coal coefficients and $CO_2$ emission factors consumed by the logistics industry.

| Energy Type | Raw Coal | Gasoline | Kerosene | Diesel | Fuel Oil | Liquefied Petroleum Gas | Natural Gas |
|---|---|---|---|---|---|---|---|
| Conversion factor (kg standard coal/kg) | 0.7143 | 1.4714 | 1.4714 | 1.4571 | 1.4286 | 1.7143 | 1.3300 |
| Average low-level heat generation (kJ/kg) | 20,908 | 43,070 | 43,070 | 42,652 | 41,816 | 50,719 | 35,605 |
| Carbon content (kg/GJ) | 26.8 | 18.9 | 19.5 | 20.2 | 21.1 | 17.2 | 15.3 |
| Carbon oxidation factor | 1 | 1 | 1 | 1 | 1 | 1 | 1 |
| $CO_2$ emission factor (kg$CO_2$/kg(m$^3$)) | 2.0553 | 2.9848 | 3.0795 | 3.1605 | 3.2366 | 3.1663 | 1.9963 |

Note: Data from China Statistical Yearbook, China Energy Statistical Yearbook, and 2006 IPCC Guidelines for National Greenhouse Gas Inventories.

In Equation (5), *E* denotes the consumption of energy, *CF* is the average low-level heat generation, *CC* is the carbon content per unit calorific value, *COF* is the carbon oxidation factor of energy, and *i* denotes the *i*th energy source.

### 3.2.2. fsQCA Model Variable Selection

The logistics industry, as a basic and supportive service industry, is involved in a wide range of tasks and is associated with many factors that can have an impact on it. On the basis of combing the related literature [12], this paper takes nine provinces in the Yellow River Basin as sample cases and refers to the practice of Lu Meili [23] to select the specific indicators of government support, economic level, scientific and technological innovation,

and degree of openness. Meanwhile, the ratio of the added value of the logistics industry to GDP in each province is selected to measure industrial agglomeration by drawing on the practice of Tang Zhe et al. [30], and the ratio of carbon dioxide emissions to the added value of the logistics industry in each province is used to characterize environmental regulation. The above six factors influencing the efficiency of the logistics industry in the Yellow River Basin are used as conditional variables in the fsQCA analysis, and the results of measuring the efficiency of the logistics industry in the Yellow River Basin of China (*ML* value) are used as outcome variables. The relevant data are mainly from China Statistical Yearbook and China Logistics Statistical Yearbook, and the specific variable descriptions are shown in Table 3.

**Table 3.** Selection of fsQCA variables.

| Variables | | Variable Symbol | Variable Description | Unit |
|---|---|---|---|---|
| Result Variables | Efficiency Value | *TFP* | Efficiency value of logistics industry by province in 2020 (*ML* value) | |
| Condition Variables | Government Support | *GOV* | Ratio of logistics-industry expenditure to total fiscal expenditure by province | % |
| | Economic Level | *PGDP* | GDP per capita by province | RMB/person |
| | Industrial Agglomeration | *IG* | Ratio of the value added of logistics industry to GDP by province | % |
| | Science and Technology Innovation | *RD* | Ratio of R&D expenditure to GDP by province | % |
| | Openness | *OPEN* | Ratio of total import and export to GDP by province | % |
| | Environmental Regulation | *ER* | Ratio of $CO_2$ emissions to value added of logistics industry by province | Million tons/RMB billion |

## 4. Empirical Results and Analysis

### 4.1. Analysis of the Results of Logistics-Industry Efficiency Measurement

#### 4.1.1. Static Analysis

Using the super-efficient SBM model based on non-expected output and MAXDEA 8.0 software, the efficiency values of the logistics industry in nine provinces in the Yellow River Basin from 2011 to 2020 are measured statically. According to the measurement results, the trend of the annual average value change of the logistics-industry efficiency for the whole region, upstream, middle, and downstream is shown in Figure 1, and the efficiency values of the logistics industry for the nine provinces in the Yellow River Basin from 2011 to 2020 are shown in Table 4.

**Table 4.** Efficiency values of the logistics industry in the nine provinces of the Yellow River Basin, 2011–2020.

| | Qinghai | Sichuan | Gansu | Ningxia | Neimenggu | Shaanxi | Shanxi | Henan | Shandong |
|---|---|---|---|---|---|---|---|---|---|
| 2011 | 1.2005 | 0.3165 | 0.7866 | 1.4853 | 0.6572 | 0.4687 | 0.4677 | 1.3847 | 1.3091 |
| 2012 | 1.2762 | 0.2938 | 0.7568 | 1.6936 | 1.0235 | 0.5004 | 0.4545 | 1.4836 | 1.1816 |
| 2013 | 1.2350 | 0.3740 | 1.0277 | 1.7079 | 1.0347 | 0.6389 | 0.5798 | 1.2383 | 1.0974 |
| 2014 | 1.2306 | 0.3506 | 1.0394 | 1.5976 | 1.0320 | 0.6828 | 0.6428 | 1.2085 | 1.0847 |
| 2015 | 1.1723 | 0.4041 | 1.0234 | 1.5344 | 1.0295 | 0.6987 | 1.0053 | 1.1439 | 1.1114 |
| 2016 | 1.0457 | 0.3439 | 0.6947 | 1.5827 | 1.0149 | 0.7583 | 1.0323 | 1.1905 | 1.1114 |
| 2017 | 0.5529 | 0.3359 | 0.7240 | 1.8385 | 1.0663 | 0.7504 | 1.4726 | 1.2057 | 1.1386 |
| 2018 | 0.5335 | 0.3478 | 1.0051 | 2.1391 | 1.1037 | 0.6510 | 1.1434 | 1.0841 | 1.0847 |
| 2019 | 0.4840 | 0.3685 | 1.0405 | 2.0613 | 1.0273 | 0.6885 | 1.1894 | 1.1228 | 1.1013 |
| 2020 | 0.4288 | 0.3482 | 0.6167 | 2.3502 | 1.0456 | 1.0422 | 1.1468 | 0.9042 | 1.1138 |
| Average | 0.9159 | 0.3483 | 0.8715 | 1.7991 | 1.0035 | 0.6880 | 0.9135 | 1.1966 | 1.1334 |
| Sort by | 5 | 9 | 7 | 1 | 4 | 8 | 6 | 2 | 3 |

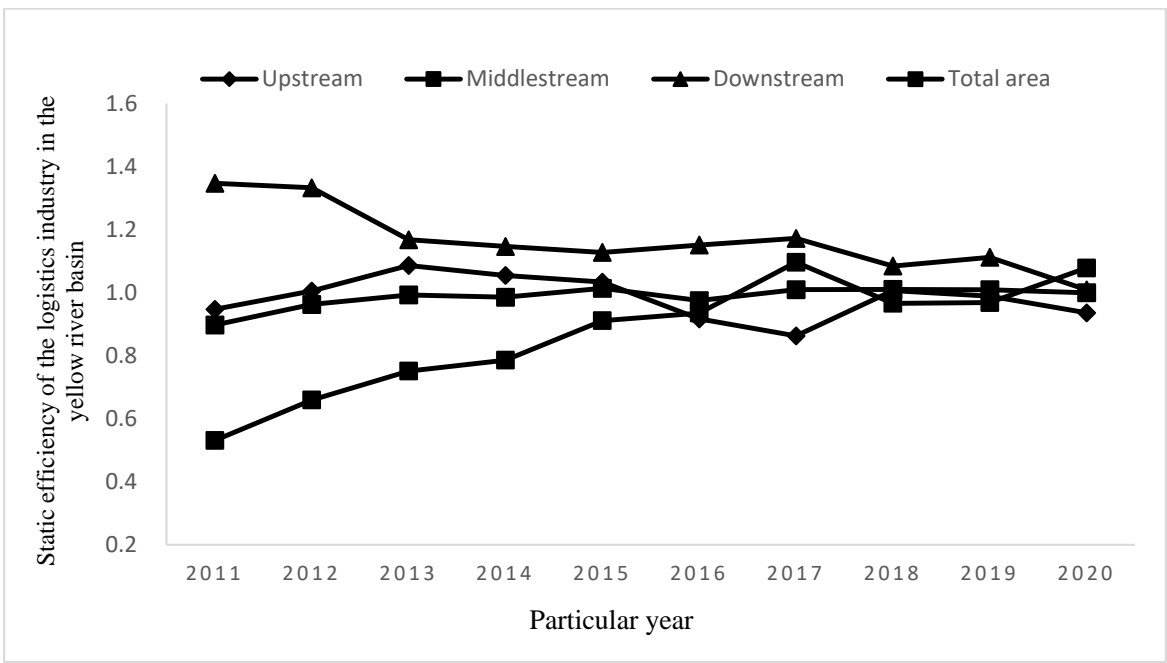

**Figure 1.** Annual average of logistics-industry efficiency in the upper, middle, and lower reaches of the Yellow River Basin from 2011 to 2020.

From Figure 1, we can see that the efficiency of the logistics industry in the Yellow River Basin generally shows a flat upward trend during the study period. However, the development differs between the upper, middle, and lower reaches of the region. The downstream is the best developed, with an average efficiency value of 1.165; the upstream is the second, with an average efficiency value of 0.984; and the midstream is the worst, with an average efficiency value of 0.868. During the study period, the trend in the downstream ranged from high to low, the trend in the midstream ranged from low to high, the trend in the upstream moved up and down below 1, and eventually all three trend lines converged to 1. This is because the logistics industry is a highly mobile industry, and the close communication and interaction between regions is conducive to learning from each other, which improved the development imbalance and gradually converged the efficiency values.

As seen in Table 4, from the average values of 2011–2020, four provinces with total factor productivity scores of the logistics industry above 1 are Ningxia, Henan, Shandong, and Inner Mongolia (in order). Among these, the highest efficiency value was obtained for Ningxia, which is similar to the results measured by other scholars [33]. No. 5, Qinghai Province, had efficiency values in 2011–2016 above 1, but the efficiency values in 2017–2020 plummeted to about 0.5, which shows that Qinghai Province's logistics facility investment has not yet produced large economic benefits. No. 6, Shanxi Province, showed a logistics-industry efficiency value that climbed from 0.47 in 2011 all the way to 1.15 in 2020—a remarkable progress. The efficiency value of Sichuan Province, which ranks last, has not reached 1 in ten years and has been at a low level. The possible reason is that Sichuan Province has a vast territory with complex terrain such as plateaus, hills, basins, and rivers, and the province can only build bridges in case of water and caves in case of mountains, making the construction of logistics infrastructure difficult. This leads to excessive investment in fixed assets in the logistics industry in Sichuan Province, which has been at the top for ten years, limiting the improvement of logistics-industry efficiency.

4.1.2. Dynamic Analysis

The Malmquist–Luenberger (*ML*) index model was applied to dynamically measure the efficiency of the logistics industry in the Yellow River Basin, and the *ML* index and decomposition term mean values of the efficiency of the logistics industry in the whole

Yellow River Basin upper, middle, and lower reaches, as well as nine provinces, from 2011 to 2020 were obtained, and the specific values and change trends are shown in Table 5 and Figure 2.

**Table 5.** *ML* index of input-output efficiency of logistics industry and its decomposition terms in nine provinces of Yellow River Basin.

| Province | ML | EC | TC | Province | ML | EC | TC |
|---|---|---|---|---|---|---|---|
| Qinghai | 0.877 | 0.907 | 1.011 | Henan | 0.996 | 0.958 | 1.038 |
| Sichuan | 1.012 | 1.018 | 1.002 | Shandong | 1.015 | 0.983 | 1.032 |
| Gansu | 0.963 | 1.006 | 1.021 | Upstream | 0.959 | 0.997 | 0.991 |
| Ningxia | 0.987 | 1.056 | 0.929 | Midstream | 1.110 | 1.100 | 1.023 |
| Neimenggu | 1.076 | 1.065 | 1.028 | Downstream | 1.005 | 0.971 | 1.035 |
| Shaanxi | 1.114 | 1.106 | 1.014 | Whole area | 1.020 | 1.025 | 1.011 |
| Shanxi | 1.141 | 1.128 | 1.028 | | | | |

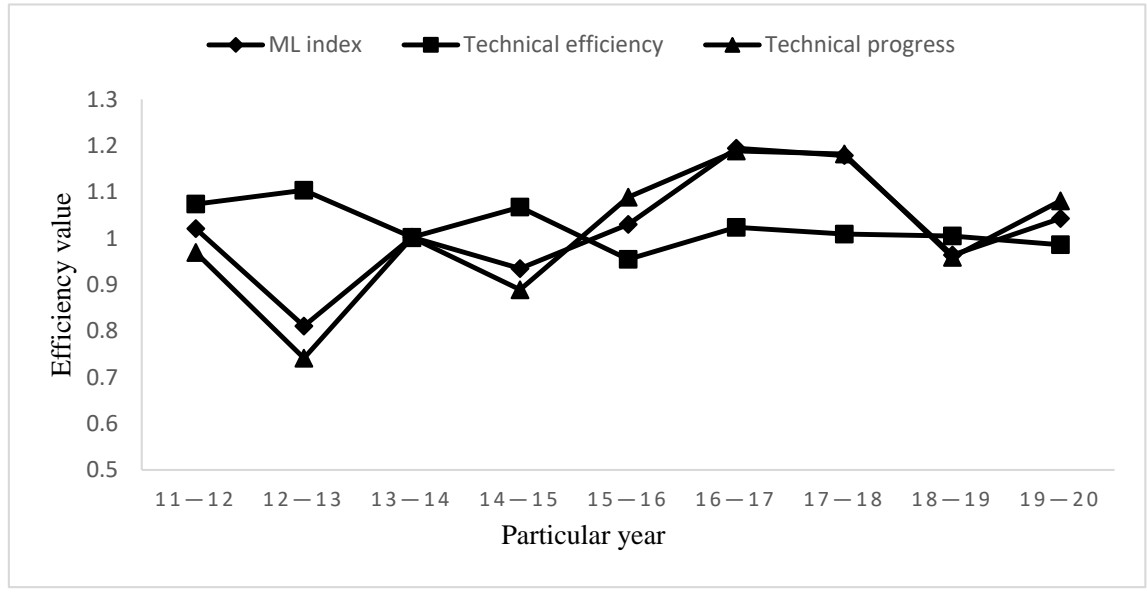

**Figure 2.** Trend of *ML* index and decomposition items of logistics industry in Yellow River Basin from 2011 to 2020.

As can be seen from Figure 2, the total factor productivity of the logistics industry in the Yellow River Basin showed a fluctuating growth in general, with an average annual growth rate of 1.99% during the period of 2011–2020. During the study period, the technical progress index (*TC*) increased by 11.16% and the technical efficiency index (*EC*) decreased by 8.76%. The rise in technical progress covered the impact caused by the decline in technical efficiency, i.e., the factor that played the main role changed from technical efficiency to technical progress. The shift of driving factors indicates that the logistics industry in the Yellow River Basin moved toward high-quality development during the decade, which is in line with the basic national conditions and policies of China.

The input-output efficiency values of the logistics industry in the nine provinces of the Yellow River Basin from 2011 to 2020 are averaged to obtain the annual average *ML* index of each province and its decomposition term, as shown in Table 5.

As can be seen from Table 5, from the regional aspect, the total factor productivity ranking of the logistics industry from high to low is midstream (1.110), downstream (1.005), and upstream (0.959), respectively. Among them, the *EC* (1.100) and *TC* (1.023) in the midstream are both greater than 1, which started a period of high development driven by two factors; the technical progress index in the downstream is the highest (1.035); and technical progress has replaced technical efficiency (0.971) as the main factor driving

growth. From the provincial side, the provinces with total factor productivity ML values greater than 1 are Shanxi, Shaanxi, Inner Mongolia, Shandong, and Sichuan (in order); the regions where technical efficiency (*EC*) is improving are Shanxi, Shaanxi, Inner Mongolia, Ningxia, Sichuan, and Gansu; and in terms of technical progress (*TC*), except for Ningxia, the remaining eight provinces show a certain upward trend. It can be seen that there is still much room for progress in the efficiency of the logistics industry in the Yellow River Basin, and the resource allocation still needs to be further optimized. In order to further investigate which factors influence the efficiency of the logistics industry in the Yellow River Basin, this paper will use the fsQCA method to conduct further analysis.

### 4.2. Variable Data Calibration

Before using the fsQCA method for analysis, the conditional and outcome variables need to be calibrated, and in this paper, referring to the study of FISS P C [34], the 0.95, 0.5, and 0.05 quartiles of the sample data were used as three calibration points, namely, "complete affiliation point = 0.95", "crossover point = 0.5" and "completely unaffiliated point = 0.05", and the anchor points of each variable were selected as shown in Table 6.

**Table 6.** Calibration points for base data for outcome and condition variables.

|  | Variable Type | Full Affiliation (0.95) | Intersections (0.5) | Total Non-Affiliation (0.05) |
|---|---|---|---|---|
| Result Variables | Efficiency Value | 1.12 | 1.01 | 0.87 |
| Condition Variables | Government Support (*GOV*) | 10.46 | 6.35 | 3.69 |
|  | Economic Level (*PGDP*) | 71,751 | 55,021 | 41,847 |
|  | Industrial Agglomeration (*IG*) | 6.32 | 4.57 | 3.23 |
|  | Science and Technology Innovation (*RD*) | 1.63 | 0.88 | 0.44 |
|  | Degree of Openness (*OPEN*) | 26.78 | 8.51 | 2.18 |
|  | Environmental Regulation (*ER*) | 3.11 | 1.39 | 1.01 |

After the calibration points were set, the Compute-Calibrate (x, n1, n2, n3) function in the fsQCA 3.0 software was used to transform the raw data of each group of variables into fuzzy-set affiliation values between 0 and 1 according to the calibration points, to obtain the calibrated data of the outcome variables and the conditional variables, as shown in Table 7. In addition, to avoid the influence of the maximum fuzzy point on the analysis results, the "0.50" point of the calibrated data is transformed into "0.501" in this paper.

**Table 7.** Data after calibration of outcome and condition variables.

| Provinces | Conditional Variable | | | | | | Outcome Variable |
|---|---|---|---|---|---|---|---|
|  | *GOVfs* | *PGDPfs* | *IGfs* | *RDfs* | *OPENfs* | *ERfs* | *TFPfs* |
| Shaanxi | 0.18 | 0.87 | 0.37 | 0.65 | 0.79 | 0.06 | 0.97 |
| Ningxia | 0.42 | 0.501 | 0.501 | 0.75 | 0.17 | 0.68 | 0.92 |
| Shanxi | 0.54 | 0.29 | 0.90 | 0.501 | 0.501 | 0.501 | 0.78 |
| Shandong | 0.03 | 0.95 | 0.97 | 0.98 | 0.98 | 0.04 | 0.56 |
| Neimenggu | 0.501 | 0.95 | 0.59 | 0.29 | 0.47 | 0.38 | 0.501 |
| Qinghai | 0.99 | 0.28 | 0.18 | 0.02 | 0.02 | 0.98 | 0.20 |
| Sichuan | 0.74 | 0.63 | 0.02 | 0.501 | 0.70 | 0.78 | 0.17 |
| Gansu | 0.63 | 0.01 | 0.49 | 0.11 | 0.12 | 0.80 | 0.11 |
| Henan | 0.08 | 0.48 | 0.76 | 0.82 | 0.69 | 0.11 | 0.02 |

### 4.3. Necessity Analysis

Necessity condition analysis is used to test whether a single condition variable (containing non-states) is necessary for the formation of the outcome variable. A factor is judged to be necessary to lead to the formation of the outcome variable if the consistency score is

greater than or equal to 0.9. If the scores are all less than 0.9, it is necessary to analyze the combination of these condition variables to explore which combination of conditions acts as sufficient to lead to the outcome variable. The coverage is used to determine the strength of the explanation of the outcome variable by the conditional variables, and a larger coverage value indicates a stronger explanation of the outcome variable by the conditional variables. If a single conditional variable is necessary, that variable will be eliminated in the results of subsequent truth-table program runs of fsQCA, which will have an impact on the analysis results; thus, necessity analysis is required. In this paper, the consistency and coverage of the condition variables were calculated and collated using fsQCA 3.0 software, and the results are shown in Table 8.

**Table 8.** Necessity analysis.

| Conditional Variables | Consistency | Coverage | Conditional Variables | Consistency | Coverage |
|---|---|---|---|---|---|
| *GOVfs* | 0.513002 | 0.527981 | *~GOVfs* | 0.763593 | 0.660532 |
| *PGDPfs* | 0.737589 | 0.629032 | *~PGDPfs* | 0.458629 | 0.480198 |
| *IGfs* | 0.718676 | 0.635983 | *~IGfs* | 0.513002 | 0.514218 |
| *RDfs* | 0.725768 | 0.664502 | *~RDfs* | 0.501182 | 0.484018 |
| *OPENfs* | 0.664303 | 0.632883 | *~OPENfs* | 0.605201 | 0.561404 |
| *ERfs* | 0.510638 | 0.498845 | *~ERfs* | 0.742317 | 0.672377 |

The results show that the consistency of each conditional variable is less than 0.9, indicating that none of the individual conditional variables is sufficient to be a necessary condition for the efficiency of the logistics industry in the Yellow River Basin. Thus, the high-level efficiency of the logistics industry in the Yellow River Basin may be the result of the combined effect of several condition variables. Therefore, it is necessary to conduct a group analysis of the condition variables to explore the path of condition combinations that can influence the high level of efficiency of the logistics industry in the Yellow River Basin.

*4.4. Configuration Path Analysis*

Cohort analysis aims to explore the effects of different combinations of multiple condition variables on the outcome variable. In this paper, the calibrated outcome variable Total Factor Productivity in Logistics (*TFPfs*) and the six conditional variables were subjected to standard analysis using fsQCA 3.0 software. As a rule, the frequency threshold is set to 1 and the consistency level is set to 0.8 for a small sample of cases, and fsQCA analysis is performed on the sample data. fsQCA solutions usually include complex, intermediate, and simple solutions. Among them, the complex solution does not contain the "logical residuals", and the solution is too complex; the simple solution contains all the "logical residuals", and the solution is too simple. The intermediate solution neutralizes the advantages of both solutions and is usually considered to be able to explain the outcome variables well. Therefore, in this paper, we chose the intermediate solution, supplemented by the parsimonious solution, to analyze and obtain the results of the high-level logistics-industry efficiency grouping path in the Yellow River Basin, as shown in Table 9. The conditions that exist both in the intermediate solution and in the simple solution are identified as the core conditions, which are the key factors used to determine the efficiency of the logistics industry in the Yellow River Basin. The conditions that exist only in the intermediate solution but not in the simple solution are identified as marginal conditions, and these conditions have relatively less impact on the efficiency of the logistics industry.

As can be seen from Table 9, the six conditional variables form five combined paths, among which the core conditions of the first and second paths are similar and can be combined as Grouping Path 1 and distinguished by Grouping Path 1a and Grouping Path 1b. The consistency indices of individual histories of these five paths are between 0.8147 and 0.8639, which are higher than the judgment value of 0.8, indicating that the result is credible as a sufficient condition for the high level of logistics-industry efficiency

in the Yellow River Basin. The original coverage refers to the percentage of the sample cases that can be explained by this combination solution, which is an overall assessment of the explanation ability of this combination solution. However, the unique coverage refers to the percentage of cases that can be explained only by this combination solution, which is an evaluation of the uniqueness of the explanation ability of this combination solution. Considering the two together can fully assess the explanatory ability of the combination solution. Generally, the overall consistency of the solution is greater than 0.8 and the overall coverage of the solution is greater than 0.5, which indicates that the results are valid. In this paper, the overall consistency is 0.8306, which indicates that the group configuration under the complete solution is attributed to the high-level logistics-industry efficiency with an 83.06% degree; the overall coverage is 0.6496, which indicates that the group cases under the complete solution explain 64.96% of the degree of high-level logistics-industry efficiency.

**Table 9.** High-level logistics-industry efficiency grouping paths in the Yellow River Basin.

| Condition Variables | Configuration | | | | |
|---|---|---|---|---|---|
| | Configuration 1a | Configuration 1b | Configuration 2 | Configuration 3 | Configuration 4 |
| Government Support | ⊗ | • | ⊗ | ● | |
| Economic Level | ● | ● | ● | • | • |
| Industrial Agglomeration | ⊗ | ⊗ | ● | • | ⊗ |
| Science and Technology Innovation | | | ● | | ⊗ |
| Openness | ● | ● | • | ⊗ | |
| Environmental Regulation | ⊗ | ⊗ | | ⊗ | ⊗ |
| Consistency | 0.8639 | 0.8147 | 0.8226 | 0.8237 | 0.8341 |
| Original Coverage | 0.5097 | 0.2946 | 0.2005 | 0.4035 | 0.3278 |
| Unique Coverage | 0.3088 | 0.0019 | 0.0326 | 0.0053 | 0.0425 |
| Overall Consistency | | | 0.8306 | | |
| Overall Coverage | | | 0.6496 | | |

Note: ●• means the condition exists, ⊗⊗ means the condition is missing; ●⊗ means the core condition, •⊗ means the edge condition; blank means not sure if the condition exists.

A cross-sectional analysis of individual condition variables reveals that economic level is present in all five histological paths, indicating that economic level plays an important role in high-level logistics-industry efficiency; however, environmental regulation is missing in four histological paths and uncertain in one histological path, indicating that less environmental regulation is more conducive to the improvement of logistics-industry efficiency in the Yellow River Basin. In the vertical analysis of the five histogram paths, five condition variables become greater than the core conditions, but the case samples covered are different; thus, it can be seen that different provinces and different environments in the Yellow River Basin have different improvement paths adapted to the high-level efficiency of the logistics industry. After collating and summarizing, the following four enhancement paths are summarized:

(1) Economic Openness Path

The economic openness path corresponds to Group 1a and Group 1b in Table 9, and the core conditions of both are economic level and openness, only differing in the marginal conditions. Group 1a shows the absence of government support, industrial agglomeration, and environmental regulation, while Group 1b shows the presence of government support and the absence of industrial agglomeration and environmental regulation. Among them, the consistency of Group 1a is 0.8639 and the original coverage is 0.5097, which can explain 50.97% of the sample cases; the consistency of Group 1b is 0.8147 and the original coverage is 0.2946, which can explain 29.46% of the sample cases. The path suggests that without considering technological innovation, if the level of logistics-industry agglomeration is low and there is a lack of government support or

weak environmental regulation, the improvement of the economic development level can influence the development of the logistics industry through different paths, such as attracting capital investment, improving infrastructure, and bringing consumer demand. This is complemented by increased openness, where foreign investment brings more advanced technology, more cutting-edge management, and a more dynamic atmosphere, leading to an increasing level of economic development. However, the improvement of the economic development level will, in turn, attract more international trade and expand the degree of openness—the two promote each other, forming a good spiral upward. Subsequently, a high level of logistics-industry efficiency will finally be achieved. The representative provinces of this path are Shaanxi Province and Shandong Province. Shaanxi Province, for example, has the highest population and strongest economy in the northwest region of China. It has assumed the important role of driving the development of the five provinces in the northwest, and the level of economic development in the northwest has been in the leading position. At the same time, due to the "One Belt and One Road" strategy of opening up the highlands, Shaanxi Province is promoting the "One Belt and One Road" international transportation and trade logistics center and "three networks and three ports" core logistics system. This has become a key driving factor for improving the efficiency of the logistics industry.

(2)   Technology-Industry-Based Path

The technology-industry-based path corresponds to Grouping 2 in Table 9, with the core conditions of economic level, industrial agglomeration, and technological innovation, and the marginal conditions of the presence of openness and the absence of government support. The consistency of Group 2 is 0.8226 and the original coverage is 0.2005, which can explain 20.05% of the sample cases. The path suggests that the linkage of economic level, industrial agglomeration, and scientific and technological innovation can also produce high levels of efficiency in the logistics industry. The huge R&D investment in science and technology innovation often needs a good economic environment and a large industrial scale for support; however, the large-scale industrial landing and agglomeration also need a developed economic level and science and technology innovation capability to drive them forward. Eventually, the strong science and technology innovation capability and the large-scale industrial clusters will, in turn, promote a significant increase in economic level. The representative province of this path is Shandong Province, which has a GDP of RMB 830.959 billion in 2022 and is one of the most developed provinces in China, ranking third in the country. At the same time, Shandong Province is also the only province with all 41 major industrial categories in the country, with a solid industrial base and strong logistics demand. According to the "Modern Logistics Network Construction Action Plan" released in June 2022, Shandong Province now has 44 5A logistics enterprises and 24 key cold-chain logistics enterprises, respectively, with the number of both ranking first in the country. On top of the developed economic level and the huge scale of the logistics industry, the ratio of R&D investment to GDP in Shandong Province has been ranked among the top nine provinces in the Yellow River Basin. Moreover, the continuous investment in science and technology innovation over the years has contributed to the continuous improvement of the level of informatization, standardization, and networking of the logistics industry in Shandong Province, which has provided strong support for the high level of efficiency and high-quality development of the logistics industry.

(3)   Government Intervention-Type Path

The government intervention-based path corresponds to Group 3 in Table 9, with the core condition of government support and the marginal conditions of economic level, the existence and openness of industrial agglomeration, and the absence of environmental regulation. The consistency of Group 3 is 0.8237 and the original coverage is 0.4035, which can explain 40.35% of the sample cases. This path shows that without considering technological innovation, even if the industrial base is relatively weak, and without the support of external capital, a certain high level of efficiency in the logistics industry can be achieved

through increased government investment and scientific planning. This path is represented by western provinces such as Qinghai and Sichuan in the upper reaches of the Yellow River. Sichuan Province, for example, as the transportation hub and economic center of southwest China, is important for promoting the implementation of the "Western Development" strategy. In August 2019, the National Development and Reform Commission issued the "Western Land and Sea New Corridor Master Plan", which specifies that the Western Land and Sea New Corridor will be basically completed by 2025. In 2022, Sichuan Province will build four national logistics hubs, which will radiate across the southwest region and reach as far as Europe through the China–European Liner to the north, integrate into the Maritime Silk Road along the Yangtze River to the east, and enter the Southeast Asia region to the south. These four hubs will become an important growth pole for the development of the national logistics level and a highly competitive international logistics hub. The logistics industry in the western region, under the support of national policies, will gradually develop for the better. Additionally, the efficiency of the logistics industry in Sichuan Province is also making great strides to reach a high level.

(4)    Non-Environmental-Regulation Path

The non-environmental-regulation path corresponds to Group 4 in Table 9, where the core condition is the absence of environmental regulation and the marginal conditions are the presence of economic level and the absence of industrial agglomeration and technological innovation. The consistency of Grouping 4 is 0.8341 and the original coverage is 0.3278, which can explain 32.78% of the sample cases. The path illustrates that in less developed regions with a weak economic base, a small logistics industry, and insufficient scientific and technological innovation, the appropriate relaxation of environmental regulations can improve the efficiency of the logistics industry. The representative province of this path is Gansu Province. From the perspective of "double carbon", the Yellow River Basin is a key area for ecological protection and high-quality development in China, but the economic development of the Yellow River Basin is relatively backward, and strict environmental regulations will hinder the improvement of logistics-industry efficiency. However, the development of the logistics industry cannot be sacrificed at the expense of the environment. Green low-carbon logistics is the future development direction of the Yellow River Basin logistics industry, which is consistent with China's "double carbon" goals and high-quality development requirements. It is believed that under the stimulation of China's economic development and long-term stability of a low-carbon policy, the efficiency of the logistics industry in the Yellow River Basin will eventually be able to eliminate environmental regulations and provide motivation to achieve the synergistic and sustainable development of economic growth, logistics efficiency, and carbon-emission reduction, which is also in line with the connotation of low-carbon economic theory.

*4.5. Robustness Tests*

A review of the current literature reveals that there are two main methods for the robustness testing of fsQCA output results: adjusting the consistency threshold and adjusting the calibration threshold. Adjusting the consistency threshold is to adjust the consistency threshold upward from 0.8 to 0.85 in the standard analysis stage [35]; adjusting the calibration threshold is to adjust the original calibration points of 0.95, 0.5, and 0.05 quantiles to 0.25, 0.5, and 0.75 quantiles in the variable data calibration stage [36]. Running the fsQCA 3.0 software, it was found that the five histogram paths remained in the results obtained by the two tests, differing only slightly in the values of consistency and coverage, with no significant weakening or strengthening of the explanatory strength. Thus, it seems that the results of the conditional histogram paths in this paper are basically robust in terms of the analysis of the efficiency of the high-level logistics industry in the Yellow River Basin.

## 5. Conclusions and Recommendations

### 5.1. Research Conclusions

Taking the Yellow River Basin of China as a sample, this paper measures the efficiency of the logistics industry in the Yellow River Basin from 2011 to 2020 using the super-efficient SBM model and Malmquist index model and reveals its spatial and temporal patterns and evolutionary trends. Subsequently, based on the overall perspective and group thinking, the influencing factors and multiple concurrent causal relationships of the high-level efficiency of the logistics industry in the Yellow River Basin are explored using the fsQCA method, and the main conclusions are as follows: ① The overall development level of the logistics-industry efficiency in the Yellow River Basin is good, but the inter-regional differences are obvious and uneven. Static analysis shows that the efficiency of the logistics industry in the Yellow River Basin as a whole shows a gentle upward trend over time, with four of the nine provinces reaching effective logistics efficiency, while the downstream and upstream logistics efficiency is higher and the midstream is lower. Dynamic analysis shows that the efficiency of the logistics industry in the Yellow River Basin as a whole shows a fluctuating growth over time, with five of the nine provinces reaching effective logistics efficiency. The midstream and downstream areas also reach effective efficiency, while the upstream logistics efficiency is lower. ② Six factors, including economic level, openness, industrial agglomeration, scientific and technological innovation, government support, and environmental regulation, all have influential effects on the efficiency of the logistics industry in the Yellow River Basin, but a single factor is not enough to be a necessary condition, and a high level of logistics-industry efficiency needs to be driven by a combination of multiple factors. The analysis found that there are four grouping paths, including: the economic-openness path, technology-industry path, government-intervention path, and the non-environmental-regulation path. Each of these four groupings has different core elements, and each region should take the path suitable for its own development according to its own actual situation.

### 5.2. Countermeasures and Recommendations

Based on the above research results, the following countermeasures are proposed from the perspective of optimizing and improving the efficiency of the logistics industry in the Yellow River Basin of China:

(1) All regions in the Yellow River Basin should give full play to their own advantages to promote the overall balanced improvement of the efficiency of the logistics industry. Both static and dynamic analyses show that the logistics development in the lower reaches of the Yellow River Basin is better, and Henan and Shandong provinces should play the role of radiation and make full use of the "diffusion effect" to drive the development of the logistics industry in the middle and lower reaches. At the same time, the upstream provinces of Ningxia, Gansu, and Qinghai, as important provinces of the "Silk Road Economic Belt", should accelerate the Western Land and Sea New Corridor and the construction of other international logistics channels; give full play to the Yellow River as a waterway transport advantage through the east and west of China; strengthen the upstream with the middle and lower reaches of the provinces of regional logistics cooperation to achieve complementary advantages; and ensure that the efficiency of the logistics industry in the Yellow River Basin will continue to be stable and developing well. In addition, they should continue to increase the construction of logistics infrastructure in coastal port cities in Shandong and node cities in the Yellow River Basin inland provinces, make full use of their logistics location advantages, lead the dissemination of benefits of the Yellow River Basin economic belt, build a "channel + hub + network" operation system, and effectively promote the logistics efficiency of the Yellow River Basin as a whole balanced improvement.

(2) Henan and Shandong should accelerate the construction of an open logistics system to promote economic development. The economic level factor is shown to exist in all

group paths, and the high level of logistics-industry efficiency needs to be supported and guaranteed by regional economic development. Shandong and Henan, in the lower reaches of the Yellow River, should increase the opening up to the outside world, vigorously develop port logistics, air logistics, and multimodal transport, and utilize Qingdao port and Zhengzhou airport as a "major infrastructure of the city and a window to the world" to improve the efficiency of logistics operations and promote energy saving, emission reduction, and carbon reduction. At the same time, they should actively participate in the international division of labor, open up to the outside world to promote economic development, create a new situation of a higher level, greater scope, and deeper opening of logistics, and continuously deepen the path of economic openness to enhance the efficiency of the logistics industry.

(3) They should focus on scientific and technological innovation to improve the efficiency of the scale. The technology-industry-based path reveals that Shaanxi and Shanxi can utilize their resource advantages, increase R&D investment, and empower the green and low-carbon development of the logistics industry with technology. Shandong Province has a developed economic level and a huge-scale logistics industry; thus, it should utilize the spatial "spillover effect" to drive Inner Mongolia, Shaanxi, Shanxi, etc., to form a complete industrial chain, supply chain, and value chain, and improve the scale efficiency. At the same time, through the deep integration of scientific and technological innovation and industrial agglomeration, they should form a scientific and technological innovation system combining industry, academia, and research to better transform scientific and technological achievements into actual products and services and to continuously open up the path of technological and industrial upgrading to improve the efficiency of the logistics industry.

(4) They should give full play to the role of government intervention while paying attention to the gradual reduction of carbon according to local conditions. Shaanxi Province can use the "One Belt and One Road" international transportation and trade logistics center and the "three networks and three ports" core logistics system to drive the reasonable planning of logistics networks in Qinghai, Gansu, and Ningxia and strengthen the construction of transportation infrastructure. Provinces in the development process should vigorously respond to the national "construction of the great Northwest" call to make full use of the "Belt and Road" construction support, to create a Western Land and Sea New Corridor. Sichuan should also accelerate the construction of the four national logistics hubs so that government intervention can enhance the effect of the maximization path. At the same time, Qinghai, Gansu, and Ningxia should take into account the real situation of the region and introduce relatively mild carbon-reduction policies according to local conditions; gradually improve carbon-reduction measures in the process of the continuous optimization of the logistics industry; employ non-environmental-regulation-type paths; and continuously improve logistics efficiency.

**Author Contributions:** Conceptualization, C.D., G.Z. and Y.W. (Yuanhao Wang); methodology and software, C.D. and Y.W. (Yuanhao Wang); writing—original draft preparation, C.D., Y.W. (Yuanhao Wang) and H.L.; writing—review and editing, C.D., Y.W. (Yongjie Wu) and H.L. All authors have read and agreed to the published version of the manuscript.

**Funding:** The project was supported by Shanxi Provincial Science and Technology Strategy Research Special Project (No. 202204031401002) and Shanxi Postgraduate Innovation Project (No. 2022Y637).

**Institutional Review Board Statement:** Not applicable.

**Informed Consent Statement:** Not applicable.

**Data Availability Statement:** Partial data openly available in a public repository. The data that support the findings of this study are openly available in https://www.stats.gov.cn/. Data available on request from the authors. Partial data that support the findings of this study are available from the author upon reasonable request.

**Acknowledgments:** The author would like to thank the editor and the anonymous referees for their helpful comments and suggestions.

**Conflicts of Interest:** The authors declare no conflict of interest.

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
