# Peer review of "Evaluation of Logistics-Industry Efficiency and Enhancement Path in China’s Yellow River Basin under Dual Carbon Targets"

_sustainability, doi:10.3390/su151712848_

Round 1
Reviewer 1 Report
Dear Authors,
I had the opportunity to read and review the manuscript entitled „Evaluation of the Efficiency of the Logistics Industry in the Yellow River Basin of China and Study of the Improvement Path”.
Based on panel data from 2011 to 2020, this study aims to propose solutions for optimizing and improving the efficiency of the logistics industry in the Yellow River Basin of China.
In my opinion, the research topic is absolutely in the scope of the journal's focus. My review below suggests some improvements.
Title: informative and consistent with the study's content.
Abstract: the abstract is clear and reasonable.
Keywords: have to be reconsidered, a couple of them are already present in the title.
1. Introduction: this section discusses the importance of improving the efficiency and reducing the carbon emissions of the logistics industry in China, especially in the Yellow River Basin, to achieve high-quality development and ecological protection. The aims of the study need to be better defined.
2. Literature Review: this section reviews the existing literature on the efficiency of the logistics industry in China and abroad, and identifies the research gap in terms of methodology, scope, and influencing factors. The Introduction and Literature review sections can be combined since the Literature review partially fulfills the role of the Introduction, e.g., by defining the research gap.
3. Research Design: I find the description of the applied models and indicators to be clear and sufficient.
4. Empirical Results and Analysis: the provided results according to the three scenarios are clear. In Table 6, please check the quartile value for Total non-affiliation.
5. Conclusions and Recommendations: I miss comparing the results of the study with those of previous studies. The proposals are supported by the research results.
It is my recommendation that this article be accepted for publication after a minor revision. There were no objectionable solutions or evaluations on my part. Study results established by the authors can be valuable both theoretically and practically.
Reviewer 2 Report
This manuscript measures the efficiency of the logistics industry in the Yellow River Basin from 2011 to 2020 using the super-efficient SBM model and Malmquist index model, and reveals its spatial and temporal patterns and evolutionary trends.The structure is complete and the diagram is clear. However, I believe that the manuscript needs to undergo significant revisions before considering publication. Please consider the following comments and make significant revisions:
1.I don't have any better suggestions for the English expression of the manuscript, but it seems that it was directly translated from Chinese, and many details need to be considered. For example, in the introduction of Chapter 1, "On October 16, 2022, the 20th Party Congress report clearly pointed out the need to accelerate the construction of a new development pattern and focus on promoting high quality development."It needs to be clearly defined as the Communist Party of China.When citing data in the manuscript, it is necessary to clarify its source (country, region, institution, year, etc.)
2. There are many explanations of existing methods in the manuscript, please simplify and emphasize the scientificity and universality of the methods proposed in the manuscript.
3.The manuscript uses data from the Yellow River Basin. Please state whether the conclusion is universal or only applicable to the Yellow River Basin. In addition.
4.The results of the manuscript analysis seem to be obvious facts. Apart from analyzing the data, such conclusions can also be found in other materials. It is recommended to delve deeper into the data behind it.
Overall, this is an interesting study, but it still requires a lot of necessary improvements.
This manuscript measures the efficiency of the logistics industry in the Yellow River Basin from 2011 to 2020 using the super-efficient SBM model and Malmquist index model, and reveals its spatial and temporal patterns and evolutionary trends.The structure is complete and the diagram is clear. However, I believe that the manuscript needs to undergo significant revisions before considering publication. Please consider the following comments and make significant revisions:
1.I don't have any better suggestions for the English expression of the manuscript, but it seems that it was directly translated from Chinese, and many details need to be considered. For example, in the introduction of Chapter 1, "On October 16, 2022, the 20th Party Congress report clearly pointed out the need to accelerate the construction of a new development pattern and focus on promoting high quality development."It needs to be clearly defined as the Communist Party of China.When citing data in the manuscript, it is necessary to clarify its source (country, region, institution, year, etc.)
2. There are many explanations of existing methods in the manuscript, please simplify and emphasize the scientificity and universality of the methods proposed in the manuscript.
3.The manuscript uses data from the Yellow River Basin. Please state whether the conclusion is universal or only applicable to the Yellow River Basin. In addition.
4.The results of the manuscript analysis seem to be obvious facts. Apart from analyzing the data, such conclusions can also be found in other materials. It is recommended to delve deeper into the data behind it.
Overall, this is an interesting study, but it still requires a lot of necessary improvements.
Author Response
请参阅附件。

Reviewer 3 Report
1.Are the concepts of “the Yellow River basin” and “the provinces through which the Yellow River flows” the same? For example, the area of the Yellow River basin in some provinces is significantly smaller than the area of the provinces. Can provincial level data characterize the logistics efficiency in the Yellow River Basin? The topic hasn’t been formulated rigorously.
2.The content of the article involves "double carbon" and “green development”. The title of "logistics industry efficiency" is not focused and specific enough.
3. Literature review expounds the efficiency of logistics industry from three aspects: methodology, scope and influencing factors.The level and logic of the literature review could be improved. For example, is there any difference between the influencing factors and the improvement path in the title? The influencing factors include positive and negative aspects, while the improvement path is mainly manifested in positive aspects.
4. The “CO2 emissions” in Table 1 should not be “Capital inputs”, but “undesirable outputs”.
5. In 3.2.2fsQCA model variable selection, please explain the rationality of the variable selection of indicators related to industrial agglomeration and environmental regulation. The authors need to add more literature to justify the choice of indicators.
6. In table 4, How to distinguish "shanxi" and "shanxi", which are two different provinces in China. “Shaanxi” appears in Table 5.
7. Is the impact of discrepancy in scale considered in data processing? It is recommended that the authors provide additional clarification in the text. For example, the order of magnitude of Economic level (PGDP) in Table 6 is significantly larger than other variables.
The language needs to be improved.
Author Response
Please refer to the annex.

Round 2
Reviewer 3 Report
Thank the authors for the work in enhancing the paper.
Minor editing of English language required.